# Improvement of In-School Physical Activity with Active School-Based Interventions to Interrupt Prolonged Sitting: A Systematic Review and Meta-Analysis

**DOI:** 10.3390/ijerph20021636

**Published:** 2023-01-16

**Authors:** Andoni Carrasco-Uribarren, Anna Ortega-Martínez, Marta Amor-Barbosa, Aida Cadellans-Arróniz, Sara Cabanillas-Barea, Maria Caridad Bagur-Calafat

**Affiliations:** 1Physiotherapy Department, Universitat Internacional de Catalunya, 08195 Barcelona, Spain; 2Physiotherapy Department, Fundació Aspace Catalunya, 08001 Barcelona, Spain

**Keywords:** physical activity, moderate-to-vigorous physical activity, sedentary behavior, sitting interruption, school, children, adolescents

## Abstract

Background: Sedentary behaviors have increased in recent years and their consequences have led the World Health Organization to make recommendations for promoting a more active lifestyle. The school environment has been defined as a key place for achieving this objective for children and adolescents. This systematic review and meta-analysis aims to analyze the effect of active-break interventions for interrupting prolonged sitting times during school-time on physical activity (PA) and sedentary behavior (SB), at school, in childhood and youth. Methods: A systematic review and meta-analysis were carried out, including clinical trials aimed at assessing the effects of interrupting prolonged sitting at school with active breaks on in-school PA and/or SB. Multimodal and static interventions were excluded. Six databases were analyzed: Medline, WOS, Cochrane Library, SPORT Discus, CINAHL and EMBASE. PA, SB; moderate-to-vigorous physical activity (MVPA) were the variables considered. Results: Nine studies were included, with a total of 2145 children between 6 and 12 years old. The heterogeneity in the duration (five–sixty min), the frequency (one–three times per-day up to three times per week), and duration (five days to three years) of the interventions was detected. The meta-analyses for in-school PA, MVPA, and SB were performed, showing a significant improvement in both PA and MVPA. Conclusions: Interrupting prolonged sitting with active-based school interventions could improve PA and MVPA levels during school time. (PROSPERO: CRD42022358933).

## 1. Introduction

Physical activity (PA) levels are well-known health indicators, particularly during childhood and youth. They are not only related to the physical, but also to the quality of life and psychological, well-being of children and adolescents [1,2]. On the contrary, sedentary lifestyles worsen them [3]. For these reasons, and due to the growth of sedentary habits during the last two decades [3], the World Health Organization (WHO) has defined daily recommendations regarding PA for moderate-to-vigorous physical activity (MVPA), for approximately 60 min per day, as well as a reduction in sedentary behaviors (SB) [4,5].

Nevertheless, recent data has shown that, overall, PA, MVPA and SB in children and adolescents differ substantially among regions and countries around the world. An example of this fact can be found when comparing the PA levels between European youth, as Northern European’s are likely to be more active and less sedentary than their Southern counterparts, in terms of both PA and MVPA levels. In addition to this, other differences have also been observed when comparing these levels regarding sex and culture. Girls and culturally diverse populations usually engage in lighter intensity behaviors compared to boys and culturally homogeneous populations [6,7,8].

Regarding age, it has been demonstrated that activity levels decrease with increasing age [8]. Whilst it could seem that adolescence would be the most critical period for the decline of physical activity, some authors agree that this decrease actually begins at the transition from early childhood to primary school (around 6–7 years old) [7,9]. Taking this into account, it is presumable that the beginning of primary school should be considered as a key period to establish interventions aimed at improving and promoting PA. This is in line with the review from Kontostoli et al. [10], who suggest that sedentary time in children and adolescents increases year by year, due to the increasing time spent engaging in screen-based activities.

According to Tassitano et al. [11], children perform approximately 27 min of MVPA during school attendance. Nevertheless, this is approximately 50% of the daily recommendations of MVPA suggested by the WHO [5]. However, the mean of MVPA performed at the school may be influenced by the academic curricula promoted by each school. Some authors have indicated that different periods of time over a school day have a direct, but varying, influence in the decrease in the PA levels through childhood and adolescence [11]. In particular, a decrease was observed during in-school PA, indicating that children and adolescents are less active at school, despite the curricular PA remaining stable [9]. Moreover, it has also been demonstrated that the environment in which they are involved when they are at school has an impact on PA levels; PA and MVPA levels are higher when children are outdoors, compared to indoors, ranging between 4 and 18.9 min/h. Conversely, the time spent in sedentary activities increases in indoor environments [11].

Considering the large number of hours that children and adolescents spend at school or high school, it is presumable that the school environment should be considered as a place to educate and improve the habit of exercising in children and adolescents, and not only to perform physical activity [11,12]. However, it is necessary to mention that actions aimed at increasing PA in a school environment depend on the academic curricula, which, in fact, depends primarily on the particular country’s legislation. Some countries have specific policies to introduce and promote PA in schools and high schools in order to approach the WHO recommendations. For example, in Spain, 44.2% of schools dedicate a minimum of 180 min per week to Physical Education classes, while 17.4% dedicate 60 min [13]. In the Netherlands, PA has a major role in the academic curricula, ranging, at least, between 120 and 180 min of PA weekly [14]. In contrast, in the USA, PA can be removed or substituted in many cases if students have to attend any missing lesson or prepare for an exam [15]. Moreover, some countries, such as Spain [16], Ireland [17] or Italy [18], have policies within the framework of the health promotion and prevention strategy, and have introduced active breaks in the school curricula, adding these active practices to Physical Education.

Previous systematic reviews have analyzed the different methodologies aimed at increasing PA and MVPA levels throughout the academic curricula, based on different approaches (multimodal interventions, active school-based programs, etc.), and their relationship with the daily WHO recommendations [19,20,21].

This systematic review and meta-analysis aims to assess the effects of interrupting prolonged sitting at school through different exercise-based interventions on in-school PA, MVPA, and SB.

## 2. Materials and Methods

### 2.1. Study Design

The present study is a systematic review and meta-analysis. The Cochrane recommendations and the Preferred Reporting Items for Systematic Reviews and Meta-Analysis (PRISMA) statement have been followed to develop the study [22]. The protocol was registered in the International Prospective Register of Systematic Reviews (PROSPERO), with the following registration number: CRD42022358933.

### 2.2. Search Strategy

A search strategy was performed in different databases, available online: Web of Science, Medline, Cochrane Library, Sport Discus, EMBASE and CINAHL. The final searching strategy was conducted until December 2021. The reference lists of relevant studies were manually searched to find other potential trials.

The search strategy was conducted by using different terms, related to the population, the intervention, and the behavior, combined and adapted to the databases, as following: (a) ‘children’ OR ‘adolescent’ OR ‘child’ OR ‘pediatrics’ OR ‘toddler’; (b) ‘interrupt’ OR ‘bout’ OR ‘break’ OR ‘break up’ OR ‘breaking up’; and (c) ‘sedentary behavior’ OR ‘risk reduction behavior’ OR ‘posture’ OR ‘rest’ OR ‘prolonged sitting’ OR ‘sitting position’. The database searching strategies are available in the Appendix A.

### 2.3. Criteria for Selection

Some inclusion and exclusion criteria were defined to identify eligible articles, regarding the type of studies, population, and interventions.

The inclusion criteria were: (a) clinical trials (cross-over studies, clinical controlled trials, and randomized clinical trials); (b) interrupting prolonged sitting with active-based PA interventions at school for the experimental group; and (c) no intervention for the control group.

The exclusion criteria were: (a) any other type of intervention (e.g., multimodal interventions or static interventions using standing desks); (b) less than one-day interventions; (c) samples including children and adolescents with disabilities unable to follow the intervention; and (d) studies not assessing in-school PA, MVPA and/or in-school SB as outcome measures.

No restrictions for publication year nor language were applied.

### 2.4. Data Extraction

The studies collected by the searching strategy were divided into two blocks and screened by two independent authors (M.A.-B. and M.C.B.-C for the first block, and A.C.-U and M.C.B.-C for the second one). In cases of discrepancies, the authors not involved in the blocks screening acted, as an arbitrator (A.C.-U for the first and M.A.-B. for the second). Discrepancies on the final eligibility were resolved by a consensus meeting.

According to their suitability, the eligible articles were examined by title, abstract and full-text. The reasons for excluding studies were recorded. The data of interest were extracted from the selected articles and, if not available, were requested from their authors.

The collected data were extracted and classified in a study characteristics table in order to provide the descriptive synthesis. The extracted data were: (1) publication year; (2) country; (3) age (mean age and/or age range); (4) sample size; (5) type of intervention; (6) measured construct (PA, SB and/or MVPA), (7) measurement tool/s; and (8) other outcomes.

### 2.5. Methodological Quality and Risk of Bias

The methodological quality was assessed using the Physiotherapy Evidence Database scale (PEDro scale), which has shown its validity by two independent authors (MC.B.-C and M.A.-B) [23,24]. In this scale, one point is awarded if the criterion is clearly satisfied. The maximum score is eleven, indicating the highest methodology quality.

The risk of bias (RoB) was assessed using the Cochrane RoB 2 tool from the Cochrane Collaboration [25] by two independent authors (A.O.-M and A.C.-U). The RoB 2 tool is a domain-based evaluation that classifies seven domains from each randomized controlled trial into ‘low’, ‘unclear’ or ‘high’ risk of bias.

In cases of disagreement on any of the scoring, a discussion was held. M.A.-B. and A.C.-U acted as arbiters for both assessments, respectively.

### 2.6. Data Synthesis and Analysis

The qualitative synthesis was described using the data obtained on Table 1.

The quantitative analysis was performed using the RevMan 5.4. software. For obtaining the results of in-school PA, MVPA and SB, three different meta-analyses were carried out. The mean difference (MD) and standard deviations (SDs) in the final scores were the data selected for the quantitative synthesis. Based on the data on the post-intervention means and SDs, the standardized mean differences (SMD) and 95% confidence intervals were calculated.

The significance value was set at *p* < 0.05. The heterogeneity between the studies was assessed, reporting both the λ^2^ test and the I² statistics. When the I² value was less than 50%, no heterogeneity was supposed, so fixed effect models were used. When the I² was higher than 50%, mixed effects models were used.

## 3. Results

### 3.1. Literature Search and Screening

The systematic literature searches yielded 3910 references after the removal of duplicates and ineligible records. Moreover, sixty-five potential studies were detected in the article reference lists. When the title and abstract were screened, 175 eligible studies remained for full-text screening. Due to different reasons, 166 references were excluded (protocols, conference papers, non-controlled trials, excluded samples of participants, setting differing from school, different measurement time frames, other interventions or studies not assessing PA, MVPA and/or SB as outcome measures). Finally, nine studies were included: eight for quantitative analysis [26,27,28,29,30,31,32,33] and one for qualitative analysis [34]. The included studies were published between 2009 and 2020 (Figure 1).

### 3.2. Characteristics of Eligible Studies

The characteristics of the studies are summarized in Table 1. Nine studies were included, involving 2145 participants between 6–12 years. The heterogeneity was detected in the sitting time interruption duration (from 5 to 60 min), the frequency (twice a week up to 3 times per-day) and in the intervention duration (from 5 days to 3 years). The outcomes reported were in-school PA (*n* = 6) [26,27,28,29,31,32], in-school MVPA (*n* = 5) [28,30,32,33,34] and in-school SB (*n* = 4) [28,30,32,33]. Accelerometers (*n* = 6) [26,28,30,32,33,34] and pedometers (*n* = 3) [27,29,31] were the measurement tools used in the included studies. Regarding the origin of the studies, they were conducted in the USA (*n* = 3) [26,27,29], Australia (*n* = 2) [32,34], Italy (*n* = 1) [28], Spain (*n* = 1) [30], Ireland (*n* = 1) [31], and The Netherlands (*n* = 1) [33].

### 3.3. Methodological Quality and Risk of Bias

Table 2 shows the methodological assessments of all of the included studies. The mean score was 3.7 points, ranging between 2 and 7. The blinding was the most frequent bias observed in the studies; only two studies masked the outcome assessor [26,33], and only one concealed the allocation [33]. The point and variability measures data were reported in all of the papers [26,27,28,29,30,31,32,33,34], but only one study applied the intention-to-treat analysis [32].

The risk of bias assessment was performed with the Cochrane risk of bias tool for the studies included in the quantitative synthesis, and it is shown next to each meta-analysis. (Figure 2, Figure 3 and Figure 4). No studies showed a “low risk” of bias, while 75% of the studies showed a “high risk” [26,27,28,29,30,31] and 25% showed an “unclear risk” [32,33]. The randomization process and the missing data, both at the baseline and/or follow-up measurements, were the most frequent items leading to a risk of bias.

### 3.4. Synthesis of the Results

From the nine studies included in this review, only one was not included in the quantitative synthesis because, although the PA data were provided at the baseline, no post-intervention data were reported [34].

#### 3.4.1. In-School Physical Activity

In-school PA was measured in six studies [26,27,28,29,31,32]. The meta-analysis revealed that interrupting prolonged sitting with active-based interventions let to a significant improvement in in-school PA (SMD = 0.46; 95% CI: 0.28, 0.64; I²: 52%) (Figure 2).

#### 3.4.2. In-School Sedentary Behavior

In-school SB was measured in four studies [28,30,32,33]. The meta-analysis revealed that interrupting prolonged sitting with active-based interventions was not enough to acquire a significant reduction in in-school SB (SMD = −0.95; 95% CI: −2.06, 0.15; I²: 98%) (Figure 3).

#### 3.4.3. In-School Moderate-to Vigorous Physical Activity

In-school MVPA was measured in five studies [28,30,32,33,34], but only four of them were included in the quantitative synthesis [28,30,32,33]. The meta-analysis showed the studied interventions produced a significant improvement in in-school MVPA (MD = 3.20; 95% CI: 3.06, 3.35; I²: 0%) (Figure 4). In the study by Watson et al. [34], the measurement of MVPA during the school day was performed using a waist-worn ActiGraph GT3-X accelerometer. No differences between the groups were found (B = 0.30; 95% CI: −0.18, 0.78).

## 4. Discussion

This systematic review and meta-analysis has been aimed to assess the effect of interrupting prolonged sitting at school with different exercise-based interventions on PA, MVPA and SB during school time. Nine studies were included, according to the eligibility criteria.

Three meta-analyses were performed, showing that interrupting prolonged sitting with active school-based interventions increase the levels of PA and MVPA at school. These interventions also seem to reduce the time spent in SB at school; however, no significant differences were found for the reduction of the latter.

The methodological quality was generally low, and the risk of bias was found to be either unclear or high. The most frequent biases were the lack of blinding, where neither the teacher who performed the intervention nor the participants could be blinded, as found by others [19,20,26,27,28,29,30,31,32,33,34]. Regarding the methodological quality, more than half of the studies included (66%) showed low quality [27,28,29,30,31,34], whereas 11% showed moderate quality [32]. Only 22% demonstrate good methodological quality [26,33]. Blinding was once again the weakest point of the studies analyzed, where both the participants and evaluators were not blinded.

A lack of consensus for describing the interrupting prolonged sitting at school interventions could be observed; we can classify these into three categories, as other authors have previously [19,20,21]. Firstly, we found short PA sessions that interrupt the academic content, also known as active breaks. Secondly, small PA sessions that included academic content, known as curriculum-focused active breaks, were found. Thirdly, we found the well-known physically active lessons, where PA is integrated into basic learning lessons, such as Mathematics. On one hand, four studies performed active breaks without related academic content [28,31,33,34]. Gallè et al. [28] conducted active breaks in the classroom, following the “AulAttiva” program, which consisted of four exercises focused on fundamental movement skills, light aerobic activity, light strength activity and gross motor coordination; van der Berg et al. [33] designed an intervention program based on moderate-to-vigorous intensity breaks using ‘Just Dance’ videos; Murtagh et al. [31] performed activity breaks that included a series of mobility, stretching and pulse-raising exercises, combined with music breaks (Bizzy Break! program); Watson et al. [34] conducted the ACTI-BREAK intervention, which consists of daily 3 × 5 min active breaks, without academic content, into the classroom routine. Three studies performed bouts of PA including academic content [27,29,30]. On the other hand, Muñoz-Parreño et al. [30] incorporated three options of curricula-focused active breaks into five subjects: high-intensity interval training (HIIT) with academic content; HIIT with emotional intelligence content; or HIIT with academic content and teamwork. Erwin et al. [27] provided teachers with activity cards with instructions for activities with basic content and/or with specific reference to health, solar safety, or multicultural issues. Mahar et al. [29] conducted the “Energizers program”, which consisted of short classroom-based PA activities to allow students to stand and move during academic instruction. Finally, two of the included studies integrated PA into academic lessons [26,32]. Donnelly et al. [26] performed physically active lessons to deliver 90 min of MVPA per week through the “Physical Activity Across the Curricula” (PAAC); Riley et al. [32] implemented the “EASY Minds” program, which consisted of adapting Mathematics lessons to ensure that students were engaged in movement-based learning.

An additional inconsistency found was the lack of consensus in the structuring of the sessions. In this sense, the interventions were heterogeneous in terms of the duration of the sitting time interruption (5–60 min), the frequency (twice a week up to 3 times per-day) and the total duration (5 days to 3 years) [26,27,28,29,30,31,32,33,34]. These differences in type and dosage of classroom-based PA interventions to interrupt prolonged sitting appear to be directly related to the heterogeneity found in the meta-analyses. Another characteristic of the included studies that may have contributed to the heterogeneity of the meta-analyses was the way in which PA, MVPA and/or SB were evaluated. The measurement instruments for in-school PA, MVPA or SB were accelerometers (*n* = 6) [26,28,30,32,33,34] and pedometers (*n* = 3) [27,29,31]. The major handicap in the use of pedometers is that they are designed to measure movements in a single axis. For example, the movement of the upper extremities, jumping, rolling, climbing or tumbling are not recorded. Consequently, pedometers provide a biased estimate of PA volume in which it is assumed that most of the activity involves locomotor movement [35,36]. In contrast to pedometers, triaxial accelerometers can record all planes and detect movements produced by children during play [35,36]. However, some activities do not involve variations in acceleration, such as isometric muscle contraction, or require greater effort than the effort accelerometers can record, such as carrying a weight load. Therefore, accelerometers are likely to underestimate the total volume of PA if these types of activities are frequently performed [36,37].

Most of the included studies used accelerometers as an intervention assessment tool. However, the procedure followed differed between them. The first difference is related to the monitoring time frame [26,28,30,32,33,34]. Although there seems to be no consensus on this issue, recordings of between 6 to 8 h with a follow-up of at least six days are recommended [38]. The other main discrepancy is related to data processing. Although there are unclear guidelines in these terms, particularly between 6 and 18 years old, it is suggested to use less than 15 s epochs. This seems to allow the detection of common discontinuous activity in children [38]. Furthermore, different algorithms have been described to discriminate PA by intensity levels [39].

The results of the meta-analyses showed that the active-breaks to interrupt prolonged sitting, recorded by accelerometers and pedometers, significantly increases the PA levels of children during school hours (SMD = 0.46; 95% CI: 0.28, 0.64; I²: 52%). In addition, this increase was also observed for MVPA (MD = 3.20; 95% CI: 3.06, 3.35; I²: 0%), which is the type of PA most highly recommended by the WHO [5]. Among the studies included in the meta-analysis, one performed a short intervention of 5 min twice a week [28], two of them performed daily sessions of 5 to 10 min [30,33], and one performed longer breaks of 3 × 60 min per week [32]. Therefore, although the results are statistically significant, the increase in MVPA during academic hours is small; however, it seems that taking these types of breaks has a greater and positive impact on the MVPA performed during the day, as shown in other studies [19]. Therefore, it seems that interrupting long sitting periods in the school environment may be one more step to reach 60 min/day of MVPA, defined as a minimum by the WHO [4,5]. According to the data from Tassitano et al. [11], and supported by the data from the present review, adding active-based activities to the academic curricula to interrupt prolonged sitting can lead to exceeding 30 min of PA per day in the school environment. In addition, it should be considered that carrying out this type of intervention at school also has an impact on children’s habits, increasing the time dedicated to perform MVPA during the day by 12.57 min [19].

Another global concern is the recently observed increase in SB in children and youth [4,5,12]. Adolescents who perform PA every day of the week reduce the possibility of being overweight in adulthood by 5% [40]. The large amount of time that children and adolescents spend at school make this environment a key place for improving and educating children in PA to prevent future diseases and to reduce SBs. However, the results of this review showed that interrupting prolonged sitting with active-based activities is not enough to reduce SBs in children and youth during the school day (SMD = −0.95; 95% CI: −2.06, 0.15; I²: 98%). Our data are similar to the revision of Neil-Sztramko et al. [12], who showed a reduction in sedentary time of 3.78 min per day (MD −3.78 min/d, 95% CI −7.80 to 0.24). Multimodal treatments have also been proposed to decrease SBs, both at school and in leisure time. This type of intervention includes awareness and education about what it is and what it means to have an inactive and sedentary way of life, and may be a crucial factor in demonstrating their effectiveness at this point [4,5].

More studies are needed to evaluate the impact of Physical Education in the curricula at school on PA levels and SBs for each country or region. In this way, PA sessions at school could provide children with information and physical skills to promote health, safety and the awareness of the benefits of a more active lifestyle [41]. It has been shown that the strategies that include time for the organization and instruction of PA combined with real practice are associated with 24% more active learning compared to usual and routine PA [42,43].

The main limitation of present study is the heterogeneity found in interrupting prolonged sitting, regarding the duration and the frequency of the interventions of the included studies, which makes it difficult to interpret the results. As a major strength, we highlight the clinical interest of the findings of this study. To our knowledge, there are no other studies that analyze the impact of classroom-based activities on meeting the WHO’s PA and SB recommendations. Our results demonstrate that the addition of active breaks are effective in increasing the level of PA during school hours, with the potential benefits that this may entail. However, it has been observed that these interventions do not produce significant differences in the reduction of SB. Other complementary interventions are needed with the aim of reducing sitting time at school.

## 5. Conclusions

Interrupting periods of sitting during school time through active-based school interventions are effective for improving physical activity levels and moderate-to-vigorous physical activity performed during school hours, but are not enough to reduce sedentary behaviors. The methods to interrupt prolonged sitting in the studies are heterogeneous, with highly variable frequencies and durations of interventions. A consensus and more homogeneous studies are necessary in order to obtain clearer conclusions.

## Figures and Tables

**Figure 1 ijerph-20-01636-f001:**
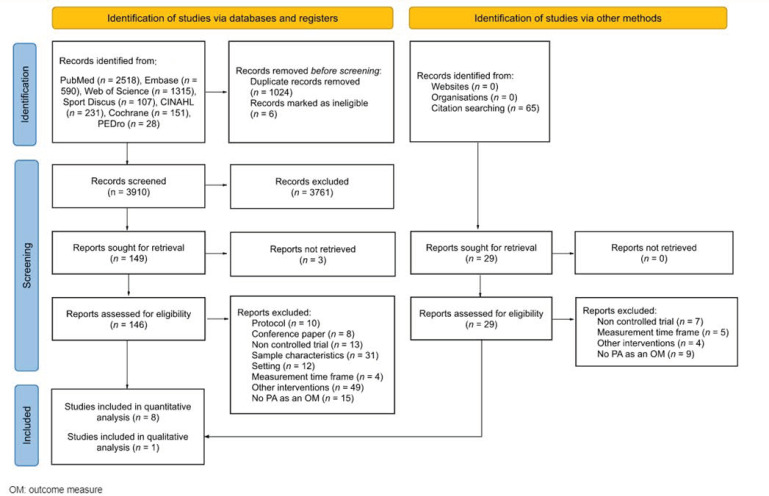
Flow diagram.

**Figure 2 ijerph-20-01636-f002:**
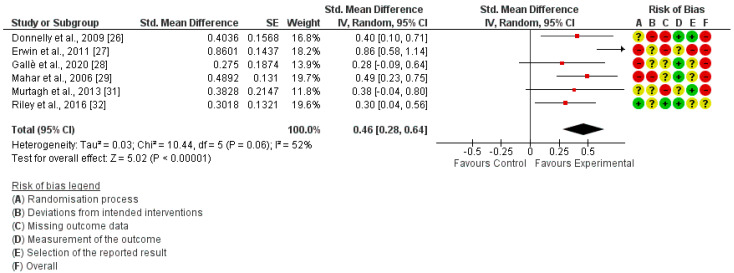
Forest plot of in−school Physical Activity.

**Figure 3 ijerph-20-01636-f003:**
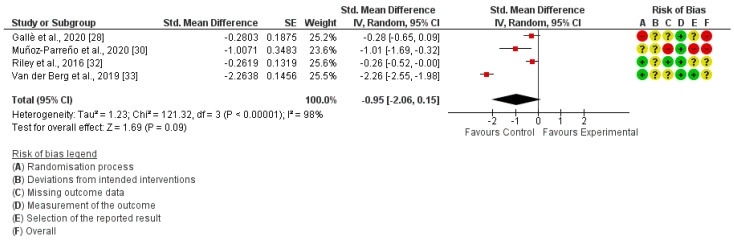
Forest plot of in−school Sedentary Behavior.

**Figure 4 ijerph-20-01636-f004:**
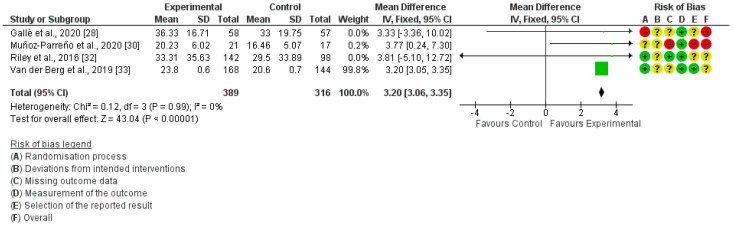
Forest plot of in−school MVPA.

**Table 1 ijerph-20-01636-t001:** Characteristics of the included studies.

Reference	Year	Country	Age		*n*	Sitting Time Interruption	Outcome Measures				
			Age Range	Mean Age (SD)			PA	MVPA	SB	Tool	Other Outcome Measures
Donnelly et al., 2009 [26]	2009	USA	6–9	3rd FCG 8,7 (0,4) 3rd FEG 8,7 (0,4) 3rd MED 8,7 (0,3) 3rd MCD 8,8 (0,4) 2nd FCG 7,8 (0,4) 2nd FEG 7,7 (0,3) 2nd MCG 7,8 (0,3) 2nd MEG 7,7 (0,4)	454	Weekly 90 min (10-min bout) for 3 years (Physical Activity Across the Curricula program)	x			Accelerometer (ActiGraph, 7163, Pensacola, FL)	Academic achievement and BMI
Erwin et al., 2011 [27]	2011	USA	8–11	10,07 (0,93)	106	At least daily 5–10 min for 8 days	x			Pedometer (Walk4Life, LS 2500, Plainfield IL)	-
Gallè et al., 2020 [28]	2020	Italy	8–9	EG 8,8 (0,4) CG 8,6 (0,3)	115	5 min twice a week for six months (AulAttiva program)	x	x	x	Accelerometer (Actigraph GT1M, Actigraph LLC, Pensacole, FL, USA)	-
Mahar et al., 2006 [29]	2016	USA	8–11	9,1 (0,9)	243	Daily 10 min for 12 weeks (Energizers program)	x			Pedometer (Yamax Digiwalker SW-200, Japan)	time on-task
Muñoz-Parreño et al., 2020 [30]	2020	Spain	9–11	10,44 (0,45)	44	Daily 5–10 min for 17 weeks		x	x	Accelerometer (ActiGraph wGT3X-BT and ActiGraph GT3X, Actigraph LLC, Pensacole, FL, USA)	-
Murtagh et al., 2013 [31]	2013	Ireland	7–12	9,3 (1,4)	90	Daily 10 min for 5 days (Bizzy Break! Program)	x			Pedometer (Yamax Digiwalker SW-200, Japan)	-
Riley et al., 2016 [32]	2016	Australia	10–12	11,13 (0,73)	240	Daily 3 × 60 min for 6 weeks (EASY Minds program)	x	x	x	Accelerometer (ActiGraph ActiGraph GT3X, Actigraph LLC, Pensacole, FL, USA)	time on-task
van der Berg et al., 2019 [33]	2019	The Netherlands	9–12	EG 10,8 (0,6) CG 10,9 (0,7)	512	Daily 10 min for 9 weeks		x	x	Accelerometer (ActiGraph ActiGraph GT3X, Actigraph LLC, Pensacole, FL, USA)	cognitive performance and aerobic fitness
Watson et al., 2018 [34]	2018	Australia	8–10	EG 9,22 (0,61) CG 9,07 (0,63)	341	Daily 3 × 5 min for 6 weeks (ACTIBREAK program)		x		Accelerometer (ActiGraph ActiGraph GT3X, Actigraph LLC, Pensacole, FL, USA)	academic achievement and on-task behavior

FCG: female control group; FEG: female experimental group; MEG: male experimental group; MCG: male control group; EG: experimental group; CG: control group; BMI: body mass index.

**Table 2 ijerph-20-01636-t002:** PEDro scale scores.

	1. Eligibility Criteria Were Specified	2. Random Allocation	3. Concealed Allocation	4. Groups Similar at Baseline	5. Participants Blinding	6. Teachers Blinding	7. Outcome Assessors Blinding	8. Less than 15% Dropouts	9. Intention-to- Treat Analysis	10. Between-Group Statistical Comparisons	11. Point Measures and Variability Data	TOTAL
Donnelly et al., 2009 [26]	1	1	0	1	0	0	1	1	0	1	1	6
Erwin et al., 2011 [27]	1	0	0	0	0	0	0	0	0	1	1	2
Gallè et al., 2020 [28]	1	0	0	0	0	0	0	1	0	1	1	3
Mahar et al., 2006 [29]	1	0	0	0	0	0	0	0	0	1	1	2
Muñoz-Parreño et al., 2020 [30]	1	0	0	1	0	0	0	0	0	1	1	3
Murtagh et al., 2013 [31]	1	1	0	0	0	0	0	0	0	0	1	2
Riley et al., 2016 [32]	0	1	0	0	0	0	0	1	1	1	1	5
van den Berg et al., 2019 [33]	1	1	1	1	0	0	1	1	0	1	1	7
Watson et al., 2018 [34]	1	1	0	0	0	0	0	1	0	0	1	3
TOTAL	8	5	1	3	0	0	2	5	1	7	9	3,7

## Data Availability

Not applicable.

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
