# Peer review of "Improvement of In-School Physical Activity with Active School-Based Interventions to Interrupt Prolonged Sitting: A Systematic Review and Meta-Analysis"

_ijerph, 2023, doi:10.3390/ijerph20021636_

Round 1
Reviewer 1 Report
Title: Active school-based interventions to interrupt prolonged sitting improve in-school physical activity: a systematic review and meta-analysis
objective: The aim of this systematic review and meta-analysis is to analyze the effect of active-break interventions for interrupting prolonged sitting time during school-time on physical activity (PA) and sedentary behavior (SB) at school in children and adolescents.
Comments to authors
Congratulations for the excellent job. You have presented an important topic, very well designed and written.
I only have a suggestion, and it is according to the discussion, it would be good if you can introduce the strengths you difusse in your research.
1. What is the main question addressed by the research?
assess the effects of interrupting prolonged sitting at school by different exercise-based interventions on in-school PA, MVPA, and SB.
2. Do you consider the topic original or relevant in the field? Does it
address a specific gap in the field?
I consider it is very important into physiotherapy field directed to children in a specific environment and to consider the physical activity since it has showed is fundamental to improve the quality of life
3. What does it add to the subject area compared with other published
material?
Study the physical therapies modalities into the school
4. What specific improvements should the authors consider regarding the
methodology? What further controls should be considered?
I consider the methodology well written and easy to understand
5. Are the conclusions consistent with the evidence and arguments presented
and do they address the main question posed?
Yes, they are were explained
6. Are the references appropriate?
I consider they are correct
7. Please include any additional comments on the tables and figures.
The tables and figures are very useful to understand the manuscript. They are very well
Reviewer 2 Report
The manuscript with a systematic review and meta-analysis about active school-based interventions to interrupt prolonged sitting improve in-school physical activity is well written. The topic is very interesting and important.
The authors discuss - from several good aspects - the advantages and disadvanteges of the nine analyzed studies, such as the different intervention durations, various time spent in additional PA and how the PA-assesments were made in each intervention, various aspects of bias etc.
One could add to the manuscript information about and discuss - the amount of increased MVPA during the school-day for the included interventions that used accelerometers. Further, a comment could be given if the amount of increased minutes in MVPA are of practical relevance among the youth. Cf your reference Amor-Barbosa et al. 2022.
Perhaps you can also add some information - in the discussion section - about the amount of increase of total PA shown (with accelerometers + pedometers).
Finally, I would like to congratulate the autors of this article on a very significant field.